# Sequential replacement of PSD95 subunits in postsynaptic supercomplexes is slowest in the cortex

Katie Morris[1], Edita Bulovaite[2], Takeshi Kaizuka[2], Sebastian Schnorrenberg[3], Candace T Adams[1,4], Noboru Komiyama[2,5,6], Lorena Mendive-Tapia[4,7], Seth GN Grant[2,5]*, Mathew H Horrocks[1,4]*

[1]EaStCHEM School of Chemistry, University of Edinburgh, Edinburgh, United Kingdom; [2]Genes to Cognition Program, Centre for Clinical Brain Sciences, University of Edinburgh, Edinburgh, United Kingdom; [3]EMBL Imaging Centre, European Molecular Biology Laboratory, Heidelberg, Germany; [4]IRR Chemistry Hub, Institute for Regeneration and Repair, University of Edinburgh, Edinburgh, United Kingdom; [5]Simons Initiative for the Developing Brain (SIDB), Centre for Discovery Brain Sciences, University of Edinburgh, Edinburgh, United Kingdom; [6]The Patrick Wild Centre for Research into Autism, Fragile X Syndrome & Intellectual Disabilities, Centre for Discovery Brain Sciences, University of Edinburgh, Edinburgh, United Kingdom; [7]Centre for Inflammation Research, University of Edinburgh, Edinburgh, United Kingdom

*For correspondence:
seth.grant@ed.ac.uk (SGNG);
mathew.horrocks@ed.ac.uk
(MHH)

Competing interest: The authors declare that no competing interests exist.

## eLife assessment

This **important** study explores how cells maintain subcellular structures in the face of constant protein turnover, focusing on neurons, whose synapses must be kept stable over long periods of time for memory storage. Using proteins from knock-in mice expressing tagged variants of the synaptic scaffold protein PSD95, nanobodies, and multiple imaging methods, there is **compelling** evidence that PSD95 proteins form complexes at synapses in which single protein copies are sequentially replaced over time. This happens at different rates in different synapse types and is slowest in areas where PSD95 lifetime is the longest and long-term memories are stored. While of general relevance to cell biology, these findings are of particular interest to neuroscientists because they support the hypothesis put forward by Francis Crick that stable synapses, and hence stable long-term memories, can be maintained in the face of short protein lifetimes by sequential replacement of individual subunits in synaptic protein complexes.

**Abstract** The concept that dimeric protein complexes in synapses can sequentially replace their subunits has been a cornerstone of Francis Crick's 1984 hypothesis, explaining how long-term memories could be maintained in the face of short protein lifetimes. However, it is unknown whether the subunits of protein complexes that mediate memory are sequentially replaced in the brain and if this process is linked to protein lifetime. We address these issues by focusing on supercomplexes assembled by the abundant postsynaptic scaffolding protein PSD95, which plays a crucial role in memory. We used single-molecule detection, super-resolution microscopy and MINFLUX to probe the molecular composition of PSD95 supercomplexes in mice carrying genetically encoded HaloTags, eGFP, and mEoS2. We found a population of PSD95-containing supercomplexes comprised of two copies of PSD95, with a dominant 12.7 nm separation. Time-stamping of PSD95 subunits in vivo revealed that each PSD95 subunit was sequentially replaced over days and weeks.

Comparison of brain regions showed subunit replacement was slowest in the cortex, where PSD95 protein lifetime is longest. Our findings reveal that protein supercomplexes within the postsynaptic density can be maintained by gradual replacement of individual subunits providing a mechanism for stable maintenance of their organization. Moreover, we extend Crick's model by suggesting that synapses with slow subunit replacement of protein supercomplexes and long-protein lifetimes are specialized for long-term memory storage and that these synapses are highly enriched in superficial layers of the cortex where long-term memories are stored.

## Introduction

Synapses in the central nervous system allow the transmission of information between neurons and are the site at which memories are formed and stored. The vast majority of synapses in the mammalian brain employ glutamate as the neurotransmitter, which is released from the presynaptic terminal on axons and diffuses onto the postsynaptic terminal on dendrites where it binds glutamate receptors (*Kandel et al., 2021*). Activating glutamate receptors triggers the biochemical changes in the postsynaptic terminal that ultimately encode memories. The proteome of the postsynaptic termini of excitatory synapses is highly complicated and comprises over 1000 proteins representing many structural and functional classes of molecules (*Sorokina et al., 2021*). Biochemical studies show that all these proteins are organized into supramolecular assemblies of complexes and supercomplexes (complexes of complexes) (*Collins et al., 2006*; *Collins and Grant, 2007*; *Fernández et al., 2017*; *Fernández et al., 2009*; *Frank and Grant, 2017*; *Frank et al., 2016*; *Husi et al., 2000*; *Husi and Grant, 2001*; *Nithianantharajah et al., 2013*; *Pocklington et al., 2006*). The best described complexes are the ionotropic glutamate receptors known as NMDA (*N*-methyl-D-aspartate) and AMPA (α-amino-3-hydroxy-5-methyl-4-isoxazolepropionic acid) receptors, which are formed from four membrane-spanning subunits that together produce a ligand-gated ion channel (*Greger et al., 2017*; *Hansen et al., 2018*). Major supercomplexes are those formed by the scaffolding protein PSD95 (*Cho et al., 1992*), which can assemble various complexes including NMDA and AMPA receptors, other ion channels, adhesion proteins, and signaling proteins (*Fernández et al., 2009*; *Frank and Grant, 2017*; *Frank et al., 2016*; *Husi et al., 2000*). Mice and humans carrying mutations in PSD95 show profound learning and memory deficits (*Migaud et al., 1998*; *Nithianantharajah et al., 2013*), indicating that the formation and function of supercomplexes is crucial for storage of information in the brain.

Although it is widely accepted that rapid biochemical changes in the postsynaptic termini of excitatory synapses underlie the initial encoding of memories, the mechanisms that control the duration of memories and rate of forgetting remain poorly understood. There has been a long-standing quest to identify molecular changes that could persist for the duration of the memory. This became a central issue in memory research in 1984 as a result of an influential article by *Crick, 1984*. He reasoned that the biochemical changes in synapses would be erased by protein turnover, and that because some memories must have a longer lifetime than synaptic proteins, he postulated that there must be molecular mechanisms that perpetuated the initial biochemical changes. Toward this, he posited that dimeric complexes of proteins may be modified during learning and gradually replace their subunits in a way that would allow the 'molecular memory' in a preexisting subunit to be transferred to the new subunit in a mixed complex of old and new subunits. As a result of this, much attention has focused on the enzymatic complex known as calcium/calmodulin-dependent protein kinase II (CamKII), which is comprised of multiple subunits that have the capacity for autophosphorylation and persistent activation (*Bayer and Schulman, 2019*; *Bhattacharyya et al., 2016*; *Hell, 2014*; *Stratton et al., 2013*). Experiments using recombinant proteins in vitro show that holoenzymes of CamKII can exchange subunits, which in principle could be a mechanism for perpetuation of molecular memories (*Bhattacharyya et al., 2020*; *Bhattacharyya et al., 2016*; *Singh and Bhalla, 2018*; *Stratton et al., 2013*). However, it is unknown if old subunits are sequentially replaced by new subunits in vivo in CamkII complexes, or indeed any other complexes in synapses.

PSD95 affords an opportunity to revisit Crick's hypotheses for several reasons. First, biochemical studies of PSD95 supercomplexes isolated from the mouse brain reveal there is a family of supercomplexes where members all appear to comprise two copies of PSD95 and different combinations of interacting proteins (*Frank and Grant, 2017*; *Frank et al., 2016*). Second, excitatory synapses show high diversity arising from the differential distribution of protein complexes (*Cizeron et al., 2020*;

*Zhu et al., 2018*) and differential lifetimes of PSD95 (*Bulovaite et al., 2022*). Those synapses with longest PSD95 lifetimes were referred to as long-protein lifetime synapses, and a brainwide analysis of their distribution showed they were most highly enriched in the cortex where long-term memories are stored. These findings raise the following questions: are the two copies of PSD95 in supercomplexes sequentially replaced, or does the dimeric complex degrade and rebuild from two new copies of PSD95? Secondly, if there is a sequential replacement of PSD95 subunits in supercomplexes, is the rate of this replacement related to the protein lifetime?

To address these questions, we have used single-molecule imaging of PSD95-containing supercomplexes isolated directly from the mouse brain. Using lines of mice that contain genetically encoded tags fused to the carboxyl terminus of endogenous PSD95 (*Broadhead et al., 2016*; *Bulovaite et al., 2022*; *Zhu et al., 2018*), we prepared protein extracts from the brain and imaged supercomplexes using total internal reflection fluorescence (TIRF) microscopy. We counted the number of PSD95 copies in each supercomplex, and found that the majority contain two units. We next measured the mean distance between the labels using MINFLUX, which has a spatial resolution of 1–5 nm, and showed that the average distance between the fluorophores is 12.7 nm. Using mice that express HaloTag fused to PSD95 (*Bulovaite et al., 2022*), we injected a fluorescent ligand into the tail vein, which efficiently labels PSD95 in brain synapses, and by extracting supercomplexes at time points after injection, we show that individual PSD95 subunits are sequentially replaced, in line with Crick's hypothesis. Finally, we ask if the rate of subunit exchange is influenced by PSD95 protein lifetime by comparing brain regions with different PSD95 lifetimes. We found that the rate of exchange is slowest in the cortex where the protein lifetime is longest. Our results, which are the first visualization of individual synaptic supercomplexes and their constituent proteins, show that synaptic scaffold proteins that play a crucial role in memory are organized as dimers in supercomplexes, and are maintained by sequential replacement of individual subunits over extended periods of time, and that this process is linked to the rate of protein turnover in the regions of the brain involved with long-term memory storage.

## Results
### Imaging individual PSD95 supercomplexes isolated from mouse brain

Single-molecule and super-resolution (SR) approaches enable the heterogeneity in molecular complexes and supercomplexes to be distinguished (*Jain et al., 2011*; *Saleeb et al., 2023*; *Szymborska et al., 2013*), and therefore provide an invaluable tool for studying synaptic molecular supercomplexes isolated from brain homogenate. Forebrains from PSD95-eGFP homozygous mice were dissected and homogenized as described (*Fernández et al., 2009*) (Materials and methods) (*Figure 1a*). The supercomplexes were subsequently diluted and immobilized on a glass coverslip, and imaged on a TIRF microscope (*Figure 1bi*) (Materials and methods). Although this technique is diffraction-limited, the number of fluorescent proteins present in each supercomplex can be quantified by the stepwise photobleaching of each fluorophore (*Dalton et al., 2016*; *Leake et al., 2006*). The number of photobleaching steps per diffraction-limited spot was determined for a population of >6000 supercomplexes across three biological repeats (*Figure 1bii*, with examples of one-step and multi-step photobleaching traces presented in *Figure 1biii*). On average, there were 1.6 PSD95 proteins per PSD95-containing supercomplex; however, taking advantage of our ability to characterize individual supercomplexes, we found that 63% contained one PSD95 protein, 24% two PSD95 proteins, and 13% more than two PSD95 proteins. Given that not all eGFP will fold correctly (typical in vitro refolding yields for GFP are 50–60% *Battistutta et al., 2000*; *Reid and Flynn, 1997*; *Ward and Bokman, 1982*), this suggests that there is an abundant population of PSD95 supercomplexes that contain two PSD95 proteins. We validated our approach using purified eGFP, demonstrating that the vast majority of eGFP was monomeric (*Figure 1—figure supplement 1*).

To further verify the PSD95 stoichiometry in supercomplexes, we generated brain homogenate from PSD95-mEos2 heterozygous mice, and following immobilization, we imaged the supercomplexes using photoactivated localization microscopy (PALM) (*Betzig et al., 2006*). Each PSD95-mEos2 molecule within the supercomplexes was localized with a mean precision of 21.8 ± 6.5 nm (mean ± SD, $n = 132,929$ localizations across three biological repeats). Fourier ring correlation (*Nieuwenhuizen et al., 2013*) revealed that the acquired images had a mean resolution of 30 ± 4 nm (mean ± SD, $n =$

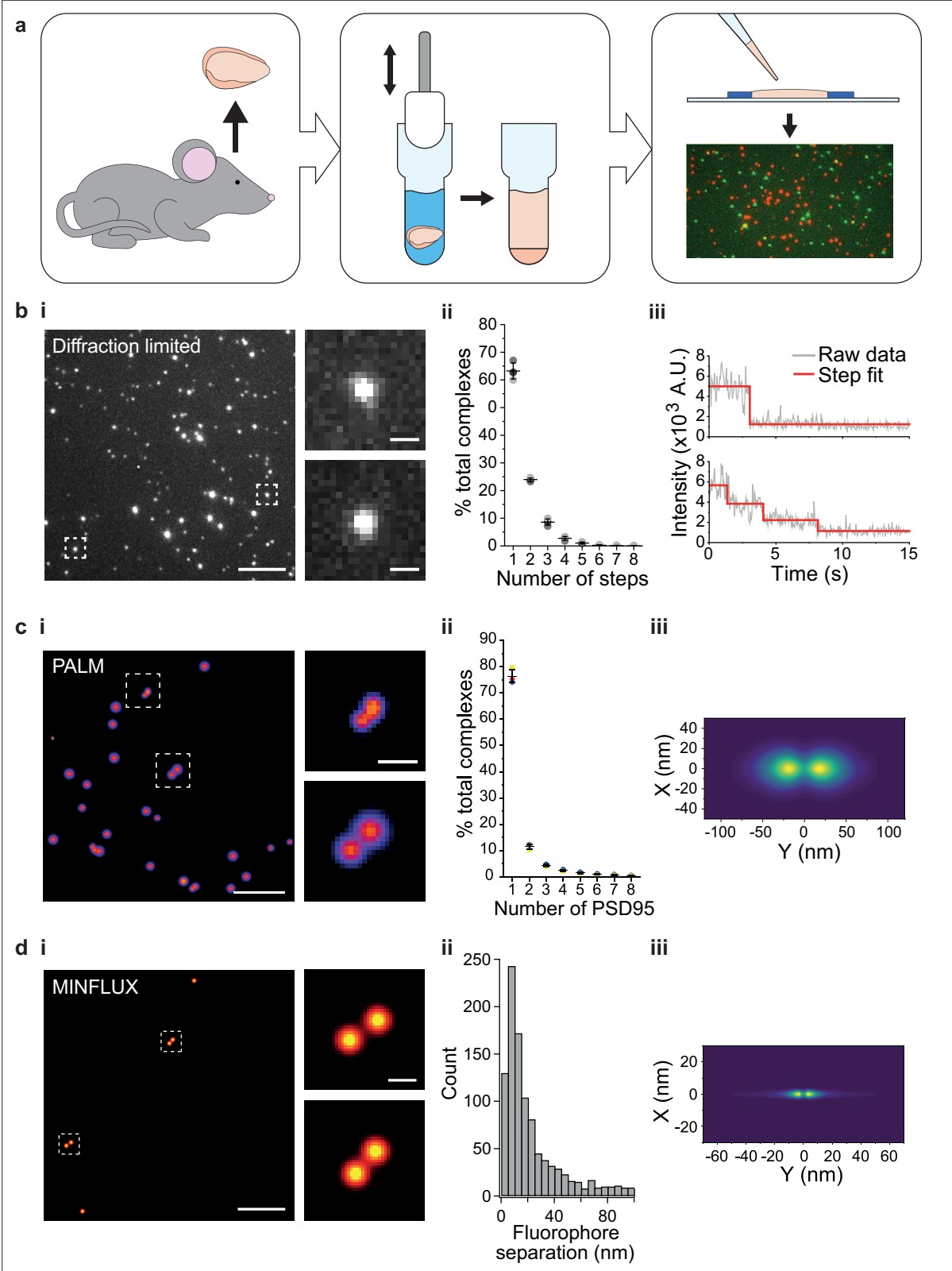

**Figure 1.** Stoichiometry and spatial arrangement of PSD95 within individual supercomplexes. (**a**) Brain containing either endogenously tagged PSD95-eGFP or PSD95-mEoS2 was extracted from the genetically modified mouse, and the forebrain was homogenized to solubilize PSD95-containing supercomplexes. PSD95 supercomplexes were immobilized on glass coverslips and imaged using single-molecule and super-resolution approaches. (**bi**) Individual PSD95-eGFP supercomplexes (boxed) were imaged using total internal reflection fluorescence (TIRF) microscopy. Scale bar = 5 μm, inset

*Figure 1 continued on next page*

*Figure 1 continued*

scale bar = 500 nm. Photobleaching step counting revealed a distribution of PSD95 stoichiometries (**bii**) (10,178 PSD95-eGFP photobleaching steps were counted across 6461 supercomplexes). Representative intensity traces with fits shown in **biii**. (**ci**) Example photoactivated localization microscopy (PALM) images of individual PSD95 supercomplexes. Scale bar = 500 nm, inset scale bar = 50 nm. (**cii**) Subsequent analysis revealed that PSD95 exists at a range of stoichiometries within the supercomplexes (132,929 PSD95-mEoS2 molecules were detected in 82,501 individual supercomplexes). Plots show mean ± SD, *n* = 3 biological repeats. (**ciii**) Class averaging of the dimer population shows a distinct separation between the PSD95 proteins within the supercomplexes (class average of 9743 supercomplexes). (**di**) Example MINFLUX images of PSD95 supercomplexes. Scale bar = 100 nm, inset scale bar = 10 nm. (**dii**) Analysis of the supercomplexes containing two PSD95 molecules (1011 supercomplexes) showed a distribution of PSD95 separation distances. (**diii**) Class averaging of this population revealed two peaks separated by 12.7 nm.

The online version of this article includes the following figure supplement(s) for figure 1:

**Figure supplement 1.** Stoichiometry of eGFP determined using photobleaching step count analysis.

**Figure supplement 2.** Multimers of PSD95 in wild-type supercomplexes.

3 biological repeats). This enabled the spatial relationship between individual PSD95 molecules to be deduced. Examples of super-resolved supercomplexes are shown in *Figure 1ci*. The number of PSD95 molecules per supercomplex was quantified using custom-written code (Materials and methods), and the distribution of stoichiometries is shown in *Figure 1cii*. Over 130,000 PSD95-mEoS2 proteins were localized in 82,501 supercomplexes across three biological repeats. 76 ± 2% of the supercomplexes contained only one PSD95-mEoS2 protein, 12 ± 1% contained two PSD95-mEoS2 proteins, and the remainder (12 ± 2%) of the clusters contained more than two PSD95-mEoS2 proteins. Given that only half of the PSD95 proteins are fused to mEoS2 in heterozygous mice, and that not all mEoS2 will fold correctly, these results confirm the findings from the photobleaching analysis, suggesting that supercomplexes contain PSD95 protein at a range of stoichiometries, with the majority containing two or fewer. We also identified multimers in wild-type PSD95 supercomplexes using a mix of orthogonally labeled PSD95 nanobodies (*Figure 1—figure supplement 2*).

We next determined the mean distance between PSD95 molecules by class averaging the 9,743 supercomplexes that contained two PSD95-mEoS2 proteins, revealing one dominant separation with a mean distance of 37.8 nm between PSD95 molecules (*Figure 1ciii*). As this separation distance is close to the spatial resolution that we were able to achieve using PALM, we further analyzed the supercomplexes using MINFLUX, which can attain a spatial resolution of 1–5 nm. We immobilized supercomplexes from brain homogenate containing PSD95-eGFP and added a GFP-nanobody tagged with Alexa Fluor 647 (Materials and methods. Details of MINFLUX imaging sequences are provided in *Table 1*). Example MINFLUX images are shown in *Figure 1di*. We were able to detect PSD95 within the supercomplexes at a range of stoichiometries, and selected for measurement those that contained two units of PSD95 (1011 supercomplexes). Class averaging the distribution of individual distances measured between two PSD95 molecules in the supercomplexes (*Figure 1dii*) shows two clear peaks separated by 12.7 nm (*Figure 1diii*).

## Sequential replacement of PSD95 within supercomplexes

We have previously measured the rate of PSD95 turnover across the mouse brain at single-synapse resolution, demonstrating that excitatory synapses have a wide range of protein lifetimes extending from a few hours to several months (*Bulovaite et al., 2022*). These findings led us to ask whether

**Table 1.** MINFLUX iterations.

| | 1st | 2nd | 3rd | 4th | 5th |
|---|---|---|---|---|---|
| L size [nm] | 300 | 300 | 150 | 75 | 40 |
| TCP – pattern | Hexagon | Hexagon | Hexagon | Hexagon | Hexagon |
| Minimum number of collected photons | 100 | 150 | 100 | 100 | 150 |
| Laser power factor | 1x | 1x | 2x | 4x | 6x |
| TCP dwell time [ms] | 1 | 1 | 1 | 1 | 1 |
| CFR | x | 0.5 | x | 0.8 | x |
| Background threshold [kHz] | 15 | 15 | 15 | 15 | 15 |

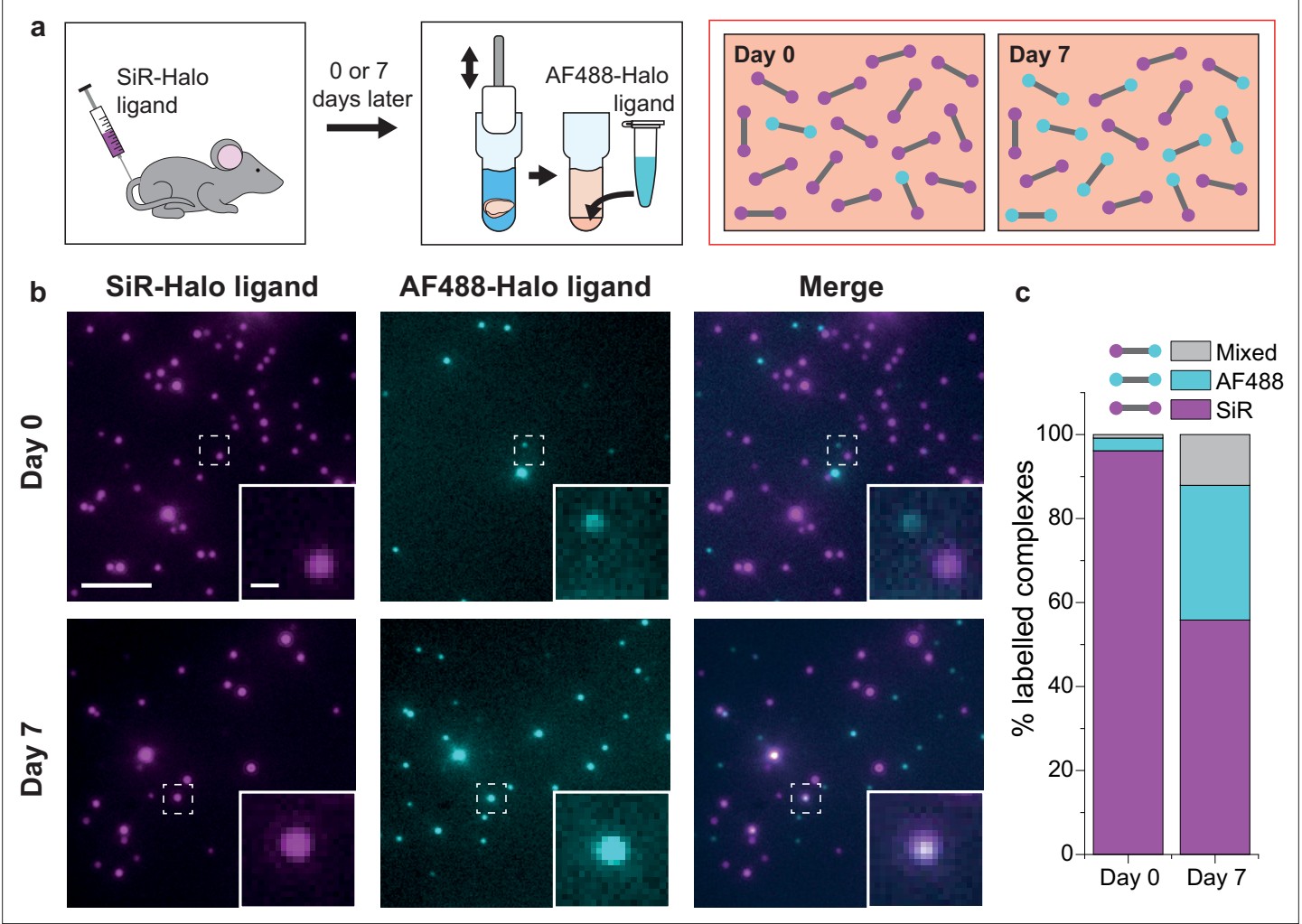

**Figure 2.** PSD95 turnover within supercomplexes in mouse brain homogenate. (**a**) PSD95-HaloTag homozygous mice were injected with SiR-Halo ligand and culled 6 hr (day-0) or 7 days (day-7) post injection. The forebrains were homogenized and post hoc stained with AF488-Halo ligand to saturate remaining binding sites. At day-0, the vast majority of PSD95 is labeled with SiR-Halo only. After 7 days of protein turnover, three populations of supercomplex are possible: SiR-Halo only, AF488-Halo only, and both fluorophores. (**b**) Images of supercomplexes from homogenate at day-0 and day-7, showing increased AF488-Halo:SiR-Halo ratios at day-7, with increased coincidence. Scale bar is 5 μm and 500 nm in the zoom. (**c**) Quantified percentages of supercomplexes labeled only with SiR-Halo, AF488-Halo, or both. At day-0, 96% of supercomplexes were labeled with SiR-Halo only, indicating saturation of PSD95-HaloTag binding sites by injection. At day-7, this had decreased to 56%, with expansion of the AF488-Halo and co-labeled populations, indicating that PSD95 protein turnover had occurred over the 7 days.

The online version of this article includes the following figure supplement(s) for figure 2:

**Figure supplement 1.** Absolute number graphs for *Figure 2c*.

protein turnover could occur within individual intact molecular supercomplexes, or if the whole super-complex needs to be replaced with new protein.

Injecting the cell- and blood–brain barrier-permeable fluorescent ligand Silicon-Rhodamine-Halo (SiR-Halo) into the tail vein of 3-month-old PSD95-HaloTag homozygous mice labels all of the PSD95-HaloTag (*Figure 2a*; *Bulovaite et al., 2022*). Because SiR-Halo forms a covalent bond with the PSD95-HaloTag, the persistence of labeling over time after injection reports whether PSD95 has been replaced in individual supercomplexes. Brain tissue was obtained at 6 hr (day-0) or 7 days (day-7) post SiR-Halo injection. The homogenized forebrain tissue from each mouse was then incubated with a second HaloTag ligand, Alexa Fluor 488-Halo (AF488-Halo), to label any new PSD95-HaloTag protein generated after the earlier SiR-Halo injection. This allowed us to quantify the levels of PSD95 protein turned over in 7 days. Supercomplexes containing only SiR-Halo represent those in which no

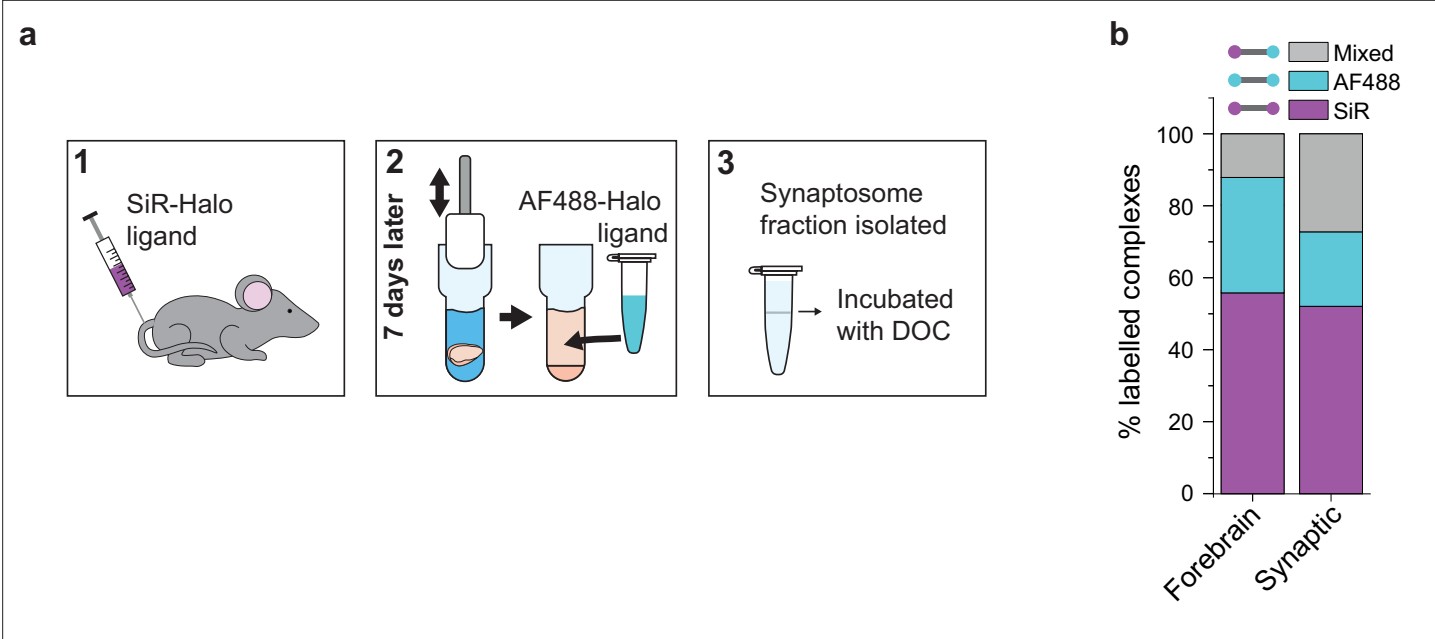

**Figure 3.** Protein turnover in synaptic and total forebrain PSD95 supercomplexes. (**a**) SiR-Halo ligand was injected into the tail vein of three 3-month-old PSD95-HaloTag knock-in mice. Seven days later, the forebrain was extracted and synaptosomes isolated. AF488-HaloTag ligand was incorporated during the homogenization process. DOC, deoxycholate. (**b**) Comparison of populations of PSD95 supercomplexes in the total forebrain and synaptic fraction.

PSD95 replacement had occurred, whereas those with only AF488-Halo are either new supercomplexes or those in which all PSD95 has been replaced. A coincident signal of SiR-Halo and AF488-Halo represents supercomplexes in which a proportion of the PSD95 has been replaced (*Figure 2a*).

Example diffraction-limited images of the labeled PSD95 supercomplexes at the two time points are shown in *Figure 2b*. Most of the supercomplexes observed at day-0 contain only SiR-Halo, with few having AF488-Halo, and negligible coincidence. At day-7, fewer SiR-Halo supercomplexes were observed, indicating that some of the PSD95 protein present at the time of injection was degraded. An increase in the number of AF488-Halo-labeled supercomplexes indicates that the degraded protein has been replaced with new, label-free protein. An increase in coincidence can be seen in the merge of the two images. These results are shown quantitatively in *Figure 2c* (see also *Figure 2—figure supplement 1*). Approximately 40% of the old PSD95 protein is replaced by new protein between day-0 and day-7. Of the 13,710 supercomplexes analyzed at day-0, 96% of the supercomplexes were labeled with only SiR-Halo, indicating that the HaloTag binding sites were saturated by injection. At day-7, 56% of the 15,391 supercomplexes analyzed were labeled with only SiR-Halo. 32% of the supercomplexes were labeled with only AF488-Halo, indicating all copies of PSD95 within these supercomplexes had been turned over in the 7-day period. Interestingly, 12% of the supercomplexes were labeled with both AF488-Halo and SiR-Halo, indicating that some supercomplexes (hereafter referred to as 'mixed supercomplexes') contain both old and new protein. This indicates that supercomplexes can exchange old copies of PSD95 for new.

## Comparison of synaptic PSD95 with total PSD95

PSD95 is synthesized in the neuronal cytosol and transported into synapses where it is concentrated in the postsynaptic density. We asked whether the exchange of PSD95 in supercomplexes differs between the synapse and the cytosol by comparing the synaptic (synaptosome) and total forebrains of mice injected with SiR-Halo at day-7 (Materials and methods). Supercomplexes were extracted from either the synaptosome fraction or the whole forebrain of SiR-Halo-injected mice and incubated with AF488-Halo and imaged using TIRF microscopy (*Figure 3a*).

Whereas the level of SiR-Halo-only supercomplexes is similar for both the total forebrain and synaptic fractions (56% and 52%, respectively), the percentage of AF488-Halo-only supercomplexes is lower in the synaptic fraction (21% versus 32% in the total forebrain homogenate) and there is a greater

fraction of mixed supercomplexes (27% versus 12% in the total forebrain homogenate) (*Figure 3b*). This suggests that while the overall rate of PSD95 turnover is similar in synapses compared with the overall forebrain, there is a greater fraction of supercomplexes that retain at least one old PSD95 protein, underlying their importance in maintaining the overall molecular state of the postsynaptic density.

## Slowest exchange of PSD95 in supercomplexes from the cortex

Because the lifetime of PSD95 is longer in cortical regions than other brain regions (*Bulovaite et al., 2022*), we hypothesized that this region may have different rates of exchange of PSD95 within supercomplexes. The brains from six 3-month-old PSD95-HaloTag mice, culled 7 days after SiR-HaloTag ligand injection, were dissected into five major regions (isocortex, hippocampus, olfactory bulb, cerebellum, and subcortex), and homogenized to extract the supercomplexes. After incubation with AF488-HaloTag ligand, to label new PSD95-Halo proteins, they were imaged using TIRF microscopy (*Figure 4a*).

In total, 60,798 supercomplexes were analyzed in the isocortex, 15,842 in the hippocampus, 3148 in the olfactory bulb, 36,126 in the cerebellum, and 28,339 in the subcortical areas. Strikingly, the isocortex contained the highest percentage of mixed supercomplexes (19 ± 3%, mean ± SD), and the olfactory bulb the lowest (4 ± 3%, mean ± SD) (*Figure 4b, c*). Correspondingly, the region with the highest percentage of new complexes was the olfactory bulb (40 ± 16%, mean ± SD), whereas the isocortex had one of the lowest percentages (27 ± 8%, mean ± SD) (*Figure 4b, c*) (statistical significances shown in *Table 2*). This pattern was also seen in 3-week-old mice (Materials and methods, *Figure 4—figure supplement 1*).

The correlation between the percentage of PSD95 proteins contained in old, new, and mixed supercomplexes in each brain region was compared with the previously published half-life of PSD95 in the same regions (Materials and methods, *Figure 4—figure supplement 2*; *Bulovaite et al., 2022*). Although there is limited correlation between the percentage of PSD95 proteins contained in old or new supercomplexes and the half-lives of PSD95 puncta in each region, there is a statistically significant correlation for mixed supercomplexes, with a Pearson's correlation test value of $R = 0.95$ at a significance level of p = 0.01.

## Discussion

The application of single-molecule detection, SR microscopy, and MINFLUX to study genetically encoded fluorescently tagged PSD95 has enabled us to reveal the structure of individual postsynaptic supercomplexes at the nanometer length scale. Although we have previously used biochemical approaches to demonstrate that, on average, two copies of PSD95 exist in supercomplexes in bulk protein extracts (*Frank and Grant, 2017*), this study represents the first direct observation of PSD95 in these megadalton supercomplexes at single-molecule resolution. Using multiple single-molecule and SR microscopy techniques to image the brain homogenate from three mouse models (PSD95-HaloTag, PSD95-eGFP, and PSD95-mEos2), we have robustly cross-validated our approach for studying the organization of PSD95 in individual supercomplexes. Unlike ensemble averaging techniques, a major advantage of our approach is the ability to observe and quantify hundreds of thousands of PSD95 molecules in tens of thousands of individual supercomplexes, allowing the stoichiometry in each one to be determined, as well as the distances between the PSD95 molecules. Furthermore, by tagging PSD95-HaloTag with specific Halo ligands in vivo, followed by imaging at different time points, our approach allows the lifetime of PSD95 in individual supercomplexes to be measured. Dissection of brain regions prior to homogenization and imaging also enables our methods to be deployed for generating maps of structure and lifetime of supercomplexes in different brain areas.

PSD95 is one of the most abundant synaptic proteins and is localized beneath the postsynaptic membrane of excitatory synapses in a structure known as the postsynaptic density. Nanoscale imaging of PSD95 in synapses using PALM and STED reveals individual synapses differ in their PSD95 content and spatial organization. For example, in the mouse hippocampus and cortex there are populations of synapses with single or multiple 40 nm 'nanoclusters' and larger contiguous structures (*Broadhead et al., 2016*; *Masch et al., 2018*). Receptors, including AMPARs have also been shown to distribute into such nanoclusters (*MacGillavry et al., 2013*) and these have been shown to align to clusters at

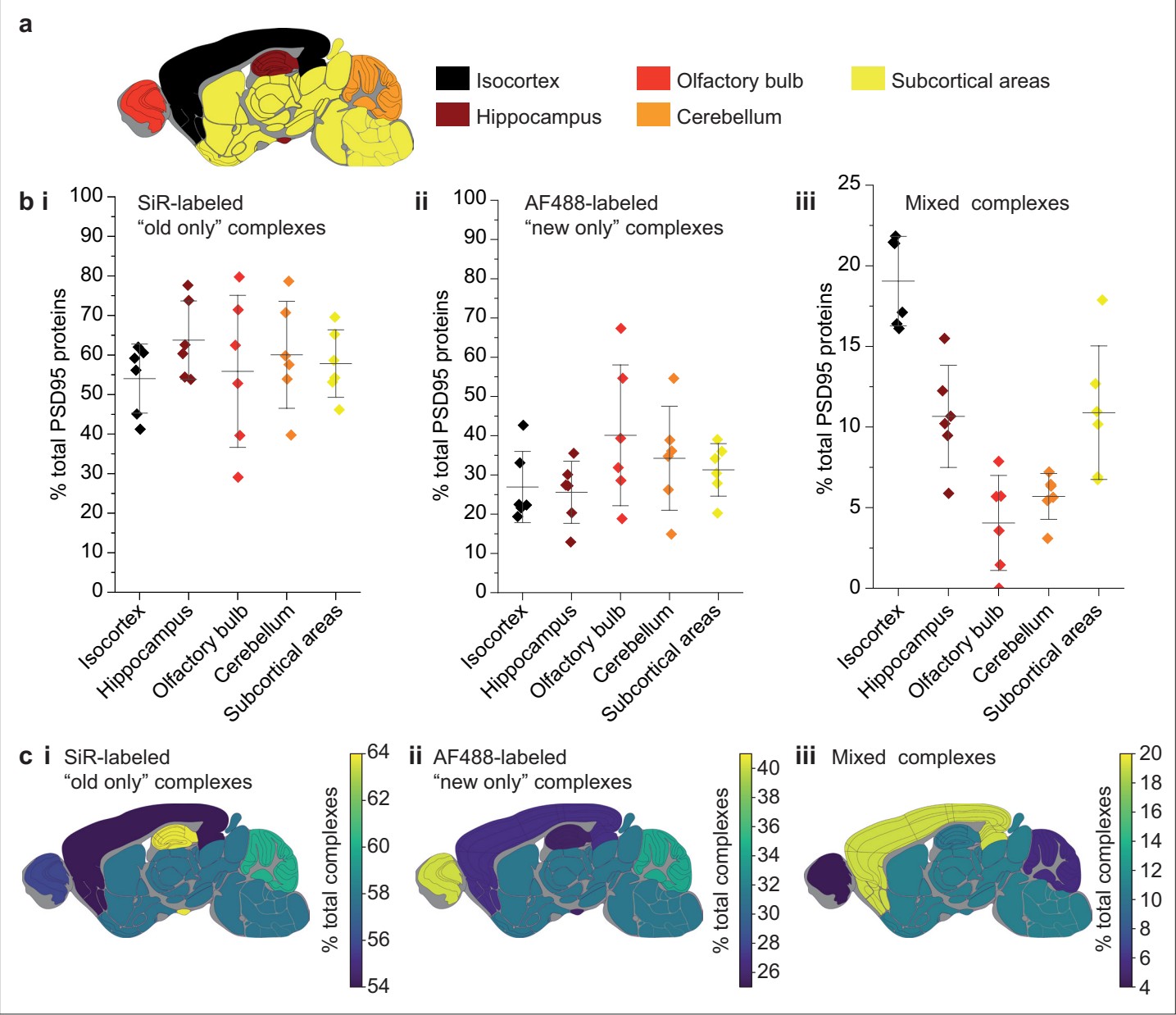

**Figure 4.** Differences in PSD95 turnover between mouse brain regions. (**a**) Mouse brains were dissected into five broad regions. (**b**) The percentage of total PSD95 imaged contained in SiR-labeled (**i**), AF488-labeled (**ii**), and mixed (**iii**) supercomplexes. Error bars show the SD of 6 biological repeats. (**c**) The percentage of SiR-labeled (**i**), AF488-labeled (**ii**), and mixed (**iii**) PSD95 supercomplexes in each region of the brain. ANOVA tests for differences between the five brain region means was carried out for plots bi, bii, and biii. The ANOVA tests returned p-values of 0.71, 0.23, and $5.5 \times 10^{-8}$, respectively. This indicates that there is no statistical significance in plots bi and bii, but there are statistical differences between some of the means in plot biii. Post hoc $t$-tests were carried out to identify the source of the significance. The p-values for comparisons between all regions are shown in **Table 2**.

The online version of this article includes the following figure supplement(s) for figure 4:

**Figure supplement 1.** Characterization of PSD95 turnover in 3-week-old mice.

**Figure supplement 2.** Comparison between the density-based PSD95 puncta half-life and old, new, and mixed proteins in supercomplexes.

the presynapse via 'nanocolumns', demonstrating their functional role (*Tang et al., 2016*). Bulk protein extracts reveal that all PSD95 is assembled into 1–3 MDa supercomplexes (*Frank and Grant, 2017*; *Frank et al., 2016*; *Husi et al., 2000*) indicating that they are the building blocks of the nanoscale structures observed with SR techniques. We now show that this synaptic heterogeneity extends to the individual supercomplex level, and demonstrate that while the majority of supercomplexes contain

**Table 2.** Post hoc *t*-tests to locate the source of significance between the means of the mixed supercomplexes in the five brain regions analyzed.

Only p-values less than 0.05 are shown. There are high levels of significance between the means of the isocortex and all other regions.

| | Isocortex | Hippocampus | Olfactory bulb | Cerebellum | Subcortex |
|---|---|---|---|---|---|
| Isocortex | | $6.4 \times 10^{-4}$ | $3.8 \times 10^{-6}$ | $1.5 \times 10^{-5}$ | $3.1 \times 10^{-3}$ |
| Hippocampus | | | $3.8 \times 10^{-3}$ | $1.0 \times 10^{-2}$ | N.S. |
| Olfactory bulb | | | | N.S. | $9.3 \times 10^{-3}$ |
| Cerebellum | | | | | N.S. |
| Subcortex | | | | | |

two or fewer PSD95 molecules, others contain more. With the enhanced resolution of MINFLUX, we were able to measure the distances between individual PSD95 molecules within the supercomplexes and find that, rather than being fixed, there is considerable variation; however, class averaging showed a dominant separation distance of 12.7 nm. This could reflect a difference in the molecular make-up of each supercomplex. Indeed, supercomplexes are composed of dozens of proteins (*Collins et al., 2006*; *Fernández et al., 2009*; *Frank and Grant, 2017*; *Husi et al., 2000*) and biochemical characterization of their composition from different brain regions show different supercomplexes populate synapses in different regions (*Frank and Grant, 2017*). Moreover, brainwide synaptome mapping of the proteins in individual synapses show synapse diversity arises from their constituent complexes and supercomplexes (*Cizeron et al., 2020*; *Tomas-Roca et al., 2022*; *Zhu et al., 2018*). Two recent studies have provided insights into the spacing between PSD95 molecules. MINFLUX revealed a mean nearest neighbor distance of approximately 7 nm (*Gürth et al., 2023*), while one-step nanoscale expansion microscopy found a preferred spacing of 8–9 nm (*Shaib et al., 2023*). These measurements, taken within the PSD, may represent distances between neighboring supercomplexes rather than within a single supercomplex, as reported here. Furthermore, in our study, the slightly larger measurements may be due to the size of the eGFP molecules (~3 nm) and the nanobodies used, which could contribute to an increased observed separation between PSD95 molecules.

The maintenance of synaptic structure at the molecular level is necessary to maintain physiological stability of brain circuits. Brainwide mapping of the spatial distribution of PSD95-expressing synapses across the lifespan shows that between 3 and 9 months of age the remarkable synapse diversity and its organization into the synaptome architecture is very stable (*Cizeron et al., 2020*). However, when we measured the lifetime of PSD95 during this age window, we found that the vast majority of PSD95 is replaced every few weeks (*Bulovaite et al., 2022*). Thus, the synaptome architecture, which is comprised of molecularly diverse synapses, is stable despite its constituent proteins being replaced. Our present findings offer an explanation for how this stability can be maintained: the protein supercomplexes which are the building blocks of the synaptome architecture are not removed and replaced in toto, but are maintained by the sequential replacement of constituents including core scaffolding proteins.

### Memory maintenance by sequential subunit replacement and protein lifetime

In the synaptome theory of behavior (*Grant, 2018*; *Zhu et al., 2018*), representations, memories, and behavioral programs are encoded in the synaptome architecture, and thus its stability is required to maintain these functional outputs. The stability conferred on the synaptome architecture by the sequential replacement of subunits in supercomplexes offers a stability mechanism. However, it is also necessary to have plasticity to learn new things and to forget unnecessary information. We propose that synapse diversity resolves this dilemma: there are some synapses that are very stable, and others that are less stable and more plastic. Our present findings indicate that the most stable synapses are those with long-protein lifetime in the cortex, and that these synapses undergo slow replacement of their supercomplex subunits. The processes of subunit replacement and protein turnover are

synergistic mechanisms that produce stability of supramolecular entities, which together with synapse diversity can result in synapses with a range of memory durations and plasticity potential.

This model fits with Crick's basic framework in that it involves dimeric proteins that sequentially exchange their subunits. Our findings extend his model in that we show a link between the protein lifetime and the rate of subunit exchange. At the time Crick made his postulate, the rate of protein turnover in the brain and synapses was unknown, and until recently, there was no evidence that synapses might differ in their protein lifetime. The surprising finding that some synapses can maintain copies of PSD95 for months draws into question Crick's assumption that protein lifetimes are much shorter than memories, because most memories are forgotten within a short period (days to weeks) in mice.

It is interesting to speculate how the exchange of PSD95 subunits could lead to the perpetuation of a molecular and behavioral memory. There are families of PSD95 supercomplexes that contain different proteins and particular interacting proteins may be well-suited to modifying the newly arrived subunit and altering its properties. The concentration of PSD95 supercomplexes is highest in the postsynaptic terminal and this environment may facilitate exchange of subunits between supercomplexes, consistent with our results. Some synapses such as the long-protein lifetime synapses that are enriched in the cortex may be better suited to the exchange process and transmission of modifications. Indeed, it may be that the environment of these synapses facilitates other complexes and supercomplexes (such as CamKII holoenzymes) to exchange and/or replace subunits and that our observations with PSD95 reflect a more widespread and general phenomenon. Our findings and approaches provide a platform for addressing these issues as well as providing a new model of memory duration.

# Materials and methods

## Preparation of mouse brain homogenates

Homogenates were prepared from dissected brains or whole mouse forebrains, as described previously (*Frank et al., 2016*). Deoxycholate (DOC) extraction buffer (1% sodium deoxycholate, 50 mM tris(hydroxymethyl)aminomethane (tris) pH 9.0, 50 mM sodium fluoride, 20 µM zinc chloride, 1 mM sodium ortho-vanadate, 2 mM 4-(2-aminoethyl)-benzene-sulfonyl fluoride, and 1 Complete Protease Inhibitor Cocktail tablet (Roche, Germany) per 50 ml) was prepared and stored on ice prior to homogenization.

For whole brain homogenization, each forebrain was added to 5 ml DOC buffer and homogenized using 20 strokes with a 5-ml capacity Teflon homogenizer. The homogenate was stored on ice for 1 hr, homogenizing again using 20 strokes at 30 min. The homogenate was centrifuged at 50,000 × $g$ for 30 min at 4°C. The supernatant was separated from the pellet and contained the PSD95 supercomplexes.

For dissected brain homogenization, each dissected brain region was added to 0.6 ml DOC buffer. Homogenization and centrifugation were performed as above for whole brain specimens, except that pellet pestles (Fisher Scientific) were used instead of a Teflon homogenizer to homogenize the tissue.

## Preparation of synaptosome fractions

Adult (5-month-old) homozygous PSD95-HaloTag mice were injected with HaloTag ligand conjugated with SiR as described previously (*Bulovaite et al., 2022*). Seven days after injection, the mice were sacrificed with cervical dislocation and the forebrain was dissected, briefly rinsed with ice-cold phosphate-buffered saline (PBS), frozen with liquid nitrogen, and stored at −80°C before use. For synaptosome preparation, homogenization of tissue was performed by 12 strokes with Teflon-glass homogenizer in Homogenization buffer (0.32 M sucrose, 1 mM 4-(2-hydroxyethyl)-1-piperazineethanesulfonic acid (HEPES) pH 7.4, and Complete EDTA-free Protease Inhibitor Cocktail (Merck)). Brain homogenate was centrifuged at 1400 × $g$ for 10 min at 4°C to obtain the pellet and the supernatant fraction. The pellet fraction was resuspended in Homogenization buffer with three strokes of the homogenizer and centrifuged at 700 × $g$ for 10 min at 4°C. The supernatant of the first and second centrifugation was pooled as an S1 fraction and subjected to centrifugation at 14,000 × $g$ for 10 min at 4°C. The resulting pellet was resuspended with Homogenization buffer (P2 fraction) and centrifuged in a sucrose density gradient (0.85/1.0/1.2 M) for 2 hr at 82,500 × $g$. The fraction between 1.0 and 1.2 M was collected and used as synaptosome.

## Dissection of mouse brain regions

Mice were culled by cervical dislocation and decapitated. The brains were removed from the skull and quickly washed with ice-cold PBS. Brains were dissected on ice, on a plastic plate covered with 3 MM filter paper soaked in ice-cold PBS. The cerebellum and olfactory bulbs were removed first, then the hippocampus and cortex were isolated from the rest of the brain. Brain samples were snap frozen in liquid nitrogen and stored at −80°C.

## Protein turnover measurements

For the protein turnover experiments, homozygous PSD95[HaloTag/HaloTag] knock-in mice were injected with 200 µl 1.5 mM SiR-Halo ligand diluted in saline and Pluronic F-127 (20% in dimethyl sulfoxide (DMSO)). As a control, some injected mice were culled 6 hr post injection to provide a 0 day time point with maximum saturation. The remainder of the mice were culled 7 days post injection. Their brains were extracted and prepared as forebrain or region-specific homogenates as described above. Prior to imaging, the turnover homogenates were incubated with 10 µM AF488-Halo ligand in a 1:1 volumetric ratio for 1 hr at 4°C.

## TIRF microscopy

All diffraction-limited and PALM experiments were conducted on a home-built TIRF microscope described previously (*de Moliner et al., 2023*). Briefly, collimated laser light at 405 nm (Cobolt MLD 405-250 Diode Laser System, Cobalt, Sweden), 488 nm (Cobolt Fandango-300 DPSS Laser System, Cobalt, Sweden), 561 nm (Cobolt DPL561-100 DPSS Laser System, Cobalt, Sweden), and 638 nm (Cobolt MLD Series 638-140 Diode Laser System, Cobolt AB, Solna, Sweden) was aligned and directed parallel to the optical axis at the edge of a 1.49 NA TIRF objective (CFI Apochromat TIRF 60XC Oil), mounted on an inverted Nikon TI2 microscope. The microscope was fitted with a perfect focus system to auto-correct the z-stage drift during imaging. Fluorescence collected by the same objective was separated from the returning TIR beam by a dichroic mirror Di01-R405/488/561/635 (Semrock, Rochester, NY, USA), and was passed through appropriate filters (488 nm: BLP01-488R-25, FF01-525/30-25; 561 nm: LP02-568RS, FF01-587/35; 638 nm: FF01-432/515/595/730-25, LP02-647RU-25, Semrock, Rochester, NY, USA). Fluorescence was then passed through a 2.5× beam expander and recorded on an EMCCD camera (Delta Evolve 512, Photometrics) operating in frame transfer mode (EMGain = 11.5 e−/ADU and 250 ADU/photon). Each pixel was 103 nm in length. Images were recorded with an exposure time of 50 ms. The microscope was automated using the open-source microscopy platform Micromanager. Borosilicate glass coverslips (20 × 20 mm, VWR International) were cleaned using an Ar plasma cleaner (Zepto, Diener) for 30 min to remove any fluorescent residues. Frame-Seal slide chambers (9 × 9 mm², Bio-Rad) were affixed to the glass to create a well in which samples (100 µl) were added. All samples in the wells were washed three times with PBS prior to imaging.

For diffraction-limited photobleaching analysis of eGFP, the neat homogenate was diluted 1:100 in PBS and irradiated and imaged with 488 nm (68 W cm⁻², 25 s), ensuring all molecules were photobleached. For diffraction-limited photobleaching analysis of SiR and AF488, whole forebrain homogenate and DOC-treated synaptosomes were diluted 1:100 in PBS. Homogenates from dissected brain regions were diluted 1:1000 to 1:10,000 in PBS. The samples were first irradiated and imaged with 638 nm light (850 W cm⁻², 25 s), followed by 488 nm light (110 W cm⁻², 25 s), ensuring all molecules were photobleached. For PALM imaging, the samples were diluted 1:100 in PBS and illuminated with 15 cycles of 405 nm (75 W cm⁻², 1 s) and 561 nm (1500 W cm⁻², 10 s) irradiation until all molecules were photobleached. In preparation for imaging, samples were incubated on the glass surface for 30 s prior to washing three times with 0.02 µm filtered PBS.

## Coincidence analysis

Coincidence analysis was carried out using a custom-written MATLAB script. Initially, all spots in the diffraction-limited images were detected using the Find Maxima function in ImageJ. The prominences used varied depending upon the fluorophores and the power of the lasers, but generally were ~500–1000. The locations of all spots in both channels were loaded into MATLAB. The distances between the *n*th spot in the first channel and all spots in the second channel were calculated. Any spots with a separation distance less than the channel offset parameter (2 pixels in this work) were classed as

being 'coincident'. Finally, the number of spots in the first channel, the number of spots in the second channel, and the number of coincident spots were output by the script.

All scripts used in this analysis are available at: https://doi.org/10.5281/zenodo.8059239; *Morris, 2023b*.

## Photobleaching analysis

Photobleaching data were collected using the TIRF microscope described above, and analyzed using a published approach (*Chappard et al., 2023*; *Choi et al., 2022*). The intensity of the emitted fluorescence from all molecules was tracked over a period of 25 s. The intensity traces from each molecule were extracted by finding spots in the first frame of the image stack using the ImageJ Find Maxima function. The intensity at each spot location was then measured in all frames of the stack using a custom-written ImageJ macro. Chung–Kennedy filtering was performed on the resulting intensity traces using a custom-written MATLAB script, with a shuttling window size of 12. The Chung–Kennedy filter (*Chung and Kennedy, 1991*) shuttles two windows along the dataset either side of each data point. The output of the Chung–Kennedy filter is a weighted average of the mean of the two windows. The weighting shifts the output value toward the mean of the window with the lowest variance such that noise is reduced, but discontinuities in the data do not become blurred during the filtering process.

To detect photobleaching steps in the filtered traces, an approximate differential of the filtered data was calculated. Peaks in the differentiated intensity traces indicated a sharp change in gradient in the intensity trace and thus the position of the photobleaching steps. A peak threshold of 75 was applied to the differentiated intensity traces to extract the location of the steps. The validity of each of the steps was determined by calculating the ratio of individual step height to the local regional variance. This calculation returned a *t*-statistic for each potential step. A *t*-threshold of 0.1 was applied to distinguish true steps from false steps. True steps have large ratios of step height to local regional variance and thus larger *t*-statistics. False steps have smaller ratios of step height to local regional variance and thus smaller *t*-statistics. Once the location of each step was known and the validity confirmed, a step function was generated as a fit to the raw step data.

All scripts used in this analysis are available at: https://doi.org/10.5281/zenodo.8059239; *Morris, 2022*.

## PALM analysis

The data were preliminarily analyzed using the Peak Fit function of the GDSC SMLM ImageJ plugin to output super-resolved localizations of the blinking fluorophores. A signal strength threshold of 20 was used, along with a precision threshold of 40 nm. Following this, the drift was corrected for using the Drift Calculator function in the GDSC SMLM ImageJ plugin. Once the localizations were extracted, a custom-written MATLAB clustering script was used. The clustering script sorted all localizations by precision from low precision to high precision. To correct for multiple blinks emanating from the same fluorophore, the script consolidated all localizations within the precision of another localization into one object. The distances between all objects were then calculated. Objects separated by less than 160 nm were grouped into clusters. The number of objects in each cluster was counted. Clusters containing one object were defined as 'monomeric', clusters containing two objects were defined as 'dimeric', etc. Information pertaining to the clusters (number of objects, *x–y* position of objects, average precision) was output as a text file.

Class averaging was performed on the class of clusters containing two objects by another custom-written MATLAB script. The script aligns all dimeric clusters parallel to the *x*-axis and centers them about the midpoint between the two objects. The objects are plotted as width = precision, and the resulting density at each *x–y* position is calculated to give the surface plot showing the class average.

All scripts used in this analysis are available at: https://doi.org/10.5281/zenodo.8059239; *Morris, 2023a*.

## MINFLUX

MINFLUX imaging was conducted on an Abberior 3D-MINFLUX microscope (Abberior Instruments, Göttingen, Germany) equipped with a 100× oil immersion objective lens (UPL SAPO100XO/1.4, Olympus, Tokyo, Japan). MINFLUX imaging of Alexa 647 was performed using a 642-nm CW excitation

laser at 22.6 µW/cm² in the first MINLUX iteration. Laser powers were measured at the position of the objective lens back focal plane using a Thorlabs PM100D power meter equipped with a S120C sensor head. Fluorescence signal from Alexa 647 was detected using two avalanche photodiodes (SPCM-AQRH-13, Excelitas Technologies, Mississauga, Canada) with a detection range of 650–685 nm for the first detector and 685–760 nm for the second detector channel (detected photons were summed). The pinhole was set to a size corresponding to 0.78 airy units for all imaging experiments. The microscope was operated by Abberior Imspector software (version 16.3.13924-m2112). The build-in stabilization system was used to minimize drift of the sample for the duration of the measurement. Therefore, scattering from 200 nm gold nanoparticles (Nanopartz, Cat# A11-200-CIT-DIH-1-10, USA) which were deposited on the coverslip surface was used as a positional reference for the active sample stabilization.

For MINFLUX imaging of PSD95-GFP, borosilicate glass coverslips (No. 1.5H, round, 24 mm diameter, product # 117640, Marienfeld, Germany) were Ar plasma cleaned for 15 min to remove any fluorescent residues. Next, coverslips were incubated with PSD95 protein lysate (1:100 in PBS) at room temperature for 1 hr. After three washing steps with PBS, coverslips were incubated with 200 nm goldparticles (Nanopartz, Cat# A11-200-CIT-DIH-1-10, USA) diluted 1:10 in PBS for 10 min. To remove goldparticles which did not attach, coverslips were washed three times with PBS and subsequently blocked using 0.5% bovine serum albumin (BSA) in PBS for 20 min at room temperature. Next, PSD95GFP was stained using an anti-GFP nanobody coupled with Alexa 647 (FluoTag-X4 anti-GFP-A647, Nanotag, Germany) diluted 1:100 in 0.5% BSA in PBS and incubated overnight at 4°C. Next day, the sample was washed three times with PBS and mounted in GLOX buffer (50 mM Tris, 10 mM NaCl, 10% glucose (wt/vol), 500 µg/ml glucose oxidase, 40 µg/ml catalase, pH 8.0) supplemented with 20 mM2-Mercaptoethylamine on cavity slides and sealed using Twinsil (Picodent, Germany). For 2D-MINFLUX imaging of PSD95-GFP, the MINFLUX sequence with five iterations was used (*Table 1*).

## Acknowledgements

J Dorrens, G Varga, E Robson for management of mouse colony and genotyping. C Davey for editing. TK was supported by a Uehara Memorial Foundation Research Fellowship, and funding from the European Union's Horizon 2020 research and innovation programme under Marie Sklodowska-Curie grant agreement No 101029343 (SYNarch). CA acknowledges support from the BBSRC EastBIO doctoral training program BB/M010996/1. LMT acknowledges funding from the Wellcome Trust Institutional Strategic Support Fund (ISSF) at the University of Edinburgh. SGNG was supported by the Simons Foundation Autism Research Initiative (529085) and a Wellcome Technology Development Grant (202932/Z/16/Z). SGNG and EB were supported by the European Research Council (ERC) under the European Union's Horizon 2020 Research and Innovation Programme (695568 SYNNOVATE; 885069 SYNAPTOME). The single-molecule instrument used in this study was funded by the UK Dementia Research Institute, UCB Biopharma, and a kind donation from Dr. Jim Love. We acknowledge the access and services provided by the Imaging Centre at the European Molecular Biology Laboratory (EMBL IC), generously supported by the Boehringer Ingelheim Foundation. For the purpose of open access, the author has applied a CC-BY public copyright licence to any Author Accepted Manuscript version arising from this submission.

## Additional information

### Funding

| Funder | Grant reference number | Author |
| --- | --- | --- |
| The Uhera Memorial Foundations | | Takeshi Kaizuka |
| Horizon 2020 Framework Programme | 101029343 | Takeshi Kaizuka |

| Funder | Grant reference number | Author |
|---|---|---|
| Biotechnology and Biological Sciences Research Council | BB/M010996/1 | Candace T Adams |
| Simons Foundation Autism Research Initiative | 529085 | Seth GN Grant |
| Wellcome Trust | 10.35802/202932 | Seth GN Grant |
| European Research Council | 695568 SYNNOVATE | Edita Bulovaite Seth GN Grant |
| European Research Council | 885069 SYNAPTOME | Edita Bulovaite Seth GN Grant |

The funders had no role in study design, data collection, and interpretation, or the decision to submit the work for publication. For the purpose of Open Access, the authors have applied a CC BY public copyright license to any Author Accepted Manuscript version arising from this submission.

## Author contributions

Katie Morris, Data curation, Formal analysis, Visualization, Methodology, Writing – original draft, Writing – review and editing; Edita Bulovaite, Takeshi Kaizuka, Data curation, Formal analysis, Visualization, Methodology, Writing – review and editing; Sebastian Schnorrenberg, Noboru Komiyama, Data curation, Visualization, Methodology, Writing – review and editing; Candace T Adams, Data curation, Visualization, Writing – review and editing; Lorena Mendive-Tapia, Data curation, Supervision, Methodology, Writing – review and editing; Seth GN Grant, Conceptualization, Supervision, Funding acquisition, Investigation, Methodology, Project administration, Writing – review and editing; Mathew H Horrocks, Conceptualization, Data curation, Formal analysis, Supervision, Funding acquisition, Investigation, Visualization, Methodology, Writing – original draft, Project administration

## Author ORCIDs

Katie Morris ⓘ https://orcid.org/0000-0002-8944-2264
Edita Bulovaite ⓘ http://orcid.org/0000-0003-2565-1371
Takeshi Kaizuka ⓘ http://orcid.org/0000-0002-5253-1896
Sebastian Schnorrenberg ⓘ https://orcid.org/0000-0002-8076-2237
Candace T Adams ⓘ https://orcid.org/0009-0001-9442-5661
Seth GN Grant ⓘ https://orcid.org/0000-0001-8732-8735
Mathew H Horrocks ⓘ https://orcid.org/0000-0001-5495-5492

## Ethics

All animal experiments conformed to the British Home Office Regulations (Animal Scientific Procedures Act 1986), local ethical approval, and NIH guidelines.

Joint Public Review https://doi.org/10.7554/eLife.99303.3.sa1
Author response https://doi.org/10.7554/eLife.99303.3.sa2

# Additional files

## Supplementary files
• MDAR checklist

## Data availability

All scripts used in this analysis are available at: https://doi.org/10.5281/zenodo.7993694 and *Morris, 2023b*.

The following dataset was generated:

| Author(s) | Year | Dataset title | Dataset URL | Database and Identifier |
|---|---|---|---|---|
| Morris K, Bulovaite E, Kaizuka T, Schnorrenberg S, Adams C, Komiyama NH, Mendive-Tapia L, Grant SGN, Horrocks MH | 2023 | s1952312/Super-resolution_NN_analysis: Super-resolution_NN_analysis-third_releasex | https://doi.org/10.5281/zenodo.7993694 | Zenodo, 10.5281/zenodo.7993694 |

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
