## [Editor Report · eLife assessment]

This **important** study explores how cells maintain subcellular structures in the face of constant protein turnover, focusing on neurons, whose synapses must be kept stable over long periods of time for memory storage. Using proteins from knock-in mice expressing tagged variants of the synaptic scaffold protein PSD95, nanobodies, and multiple imaging methods, there is **compelling** evidence that PSD95 proteins form complexes at synapses in which single protein copies are sequentially replaced over time. This happens at different rates in different synapse types and is slowest in areas where PSD95 lifetime is the longest and long-term memories are stored. While of general relevance to cell biology, these findings are of particular interest to neuroscientists because they support the hypothesis put forward by Francis Crick that stable synapses, and hence stable long-term memories, can be maintained in the face of short protein lifetimes by sequential replacement of individual subunits in synaptic protein complexes.

---

## [Referee Report · Joint Public Review]

The present study explored the principles that allow cells to maintain complex subcellular proteinaceous structures despite the limited lifetimes of the individual protein components. This is particularly critical in the case of neurons, where the size and protein composition of synapses define synaptic strength and encode memory.

PSD95 is an abundant synapse protein that acts as a scaffold in the recruitment of transmitter receptors and other signaling proteins and is required for memory formation. The authors used super-resolution microscopy to study PSD95 super-complexes isolated from the brains of mice expressing tagged PSD variants (Halo-Tag, mEos, GFP). Their results show compellingly that a large fraction (~25%) of super-complexes contains two PSD95 copies about 13 nm apart, that there is substantial turnover of PSD95 proteins in super-complexes over a period of seven days, and that ~5-20% of the super-complexes contain new and old PSD95 molecules. This percentage is higher in synaptic fractions as compared to total brain lysates, and highest in isocortex samples (~20%). These important findings support the hypothesis put forward by Crick that sequential subunit replacement gives synaptic super-complexes long lifetimes and thus aids in memory maintenance. Overall, this is a very interesting study that provides key insights into how synaptic protein complexes are formed and maintained. On the other hand, the actual role of these PSD95 super-complexes in long-term memory storage remains unknown. Specifically, a direct correlation between PSD95 stability and memory formation remains hypothetical - but the present findings indicate important new directions for studying the mechanisms that control postsynaptic protein organisation and the maintenance of postsynaptic proteinaceous substructures.

Strengths

(1) The study employed an appropriate and validated methodology.

(2) Large numbers of PSD95 super-complexes from three different mouse models were imaged and analyzed, providing adequately powered sample sizes.

(3) State-of-the-art super-resolution imaging techniques (PALM and MINFLUX) were used, providing a robust, high-quality, cross-validated analysis of PSD95 protein complexes that is useful for the community.

(4) The result that PSD95 proteins in dimeric complexes are on average 12.7 nm apart is useful and has implications for studies on the nanoscale organization of PSD95 at synapses.

(5) The finding that postsynaptic protein complexes can continue to exist while individual components are being renewed is important for our understanding of synapse maintenance and stability.

(6) The data on the turnover rate of PSD95 in super-complexes from different brain regions provide a first indication of potentially meaningful differences in the lifetime of super-complexes between brain regions.

Weaknesses

(1) The manuscript emphasizes the hypothesis that stable super-complexes, maintained through sequential replacement of subunits, might underlie the long-term storage of memory. While an interesting idea, this notion requires considerably more research. The presented experimental data are indeed consistent with this notion, but there is no evidence that these complexes are causally related to memory storage.

(2) Much of the presented work is performed on biochemically isolated protein complexes. The biochemical isolation procedures rely on physical disruption and detergents that are known to alter the composition and structure of complexes in certain cases. Thus, it remains unclear how the protein complexes described in this study relate to PSD95 complexes in intact synapses.

(3) Because not all GFP molecules mature and fold correctly in vitro and the PSD95-mEos mice used were heterozygous, the interpretation of the corresponding quantifications is not straightforward.

(4) It was not tested whether different numbers of PSD95 molecules per super-complex might contribute to different retention times of PSD95, e.g. in synaptic vs. total-forebrain super-complexes.

(5) The conclusion that the population of 'mixed' synapses is higher in the isocortex than in other brain regions is not supported by statistical analysis.

(6) The validity of conclusions regarding PSD95 degradation based on relative changes in the occurrence of SiR-Halo-positive puncta is limited.

---

## [Author Response]

The following is the authors’ response to the original reviews.

**Recommendations for the authors:**

**Reviewer #1 (Recommendations For The Authors):**
The work is well performed and thoroughly convincing.However, a few points could be improved, by adjusting the manuscript:(1) The wording of the abstract is confusing for the casual reader. The initial impression is that the 2-copy complexes contain the majority of the PSD95 copies. This is not the case, as shown in panel cii. It would be important for the authors to explain in the abstract the exact percentage of molecules found within 2-copy complexes.

We have now amended the abstract, making it clear that it’s not most of the complexes.

(2) Did the authors find a sizeable population of 2-copy complexes by investigating wild-type proteins, using nanobody labeling (Figure S2)? It would be important to quantify and discuss these data.

It was not possible to perform this analysis on the wild-type proteins. The quantification would rely on all the PSD95 molecules being bound by the antibody, which we cannot guarantee. Furthermore, the nanobody labeling would need to be stoichiometric.

(3) The authors quote the separation value of 12.7 nm throughout their text, including the abstract. This may be somewhat misleading since the authors investigate the PSD95-GFP molecules, labeled using anti-GFP nanobodies. The large size of the two GFP molecules (~3 nm), and that of the nanobodies, will influence the readout. Two groups have already reported a separation of ~7-8 nm between neighboring PSD95 molecules in synapses, using PSD95 nanobodies, to minimize the linkageerror: https://doi.org/10.1101/2022.08.03.502284 and https://doi.org/10.1101/2023.10.18.562700.The difference observed here is consistent with an effect of the additional GFP moieties; the authors should cite these works (albeit they are now only provided as biorXiv pre-prints) and should mention this discrepancy, and its potential tagging-related explanation.

We have now referenced the work and referred to this in the discussion.

(4) The authors may want to re-check the manuscript; some minor problems should be corrected, such as the mislabeling of Figure 2 and "Figure 5".

This has now been corrected.

**Reviewer #2 (Recommendations For The Authors):**
The authors suggest that the stability of the PSD95 dimeric complex correlates with memory formation. However, the turnover experiments were conducted only on three-month-old animals, which can be considered to be at a stage of lower synaptic functionality turnover. It would be appropriate to study dimer turnover during the memory formation period at three to four weeks of age, for example in comparison to the oldest mice.Alternatively, it might be interesting to study the turnover in the hippocampus following exposure to a memory test.

Whilst potentially useful, these experiments are outside of the scope of this manuscript.

It is not clear whether the different turnover identified in various brain areas is statistically significant, as apparently no statistical analysis has been conducted.

The findings were significant, and the SI table containing the p-values has been emphasized further in the manuscript.

**Reviewer #3 (Recommendations For The Authors):**
(1) In the last paragraph of the Results section, it could be made clearer what the nature is of the correlation between PSD95 half-life and mixed supercomplexes to understand how to interpret this correlation. In the discussion, it is concluded that stable synapses have long protein lifetimes and slow replacement of scaffolding proteins. However, this is based on the correlation of protein lifetime and mixed supercomplexes in the cortex, which does not provide any evidence that this relation is true in single synapses or is specific for stable synapses. To make this statement, the authors could for instance directly correlate the stoichiometry of supercomplexes with the protein lifetime and size of individual synapses.

Unfortunately, we can’t directly measure the lifetime of each complex, and so it’s only possible to compare region-to-region. In doing so, we found that there was a correlation between the protein lifetime and the “mixed” population.

(2) Some essential parts seem missing: the materials and methods and Figure 2 are not included. Also, the numbering of figures is incorrect. Both in the figure legends and the text.

This has been added.

(3) Figure 1a could contain more details of the experimental procedures. For example, it could be made clearer how PSD95 supercomplexes are isolated from brain homogenate.

This is now presents in the methods.

(4) In Figure 1c, single molecules of PSD95 are identified using PALM with a resolution of 30 nm. However, in Figure 1d it is shown that PSD95 molecules reside on average 13 nm apart, indicating that a resolution of 30 nm is not sufficient to resolve single PSD95 molecules. In addition, it would be of interest to show the distribution of fluorophore separation (assessed with MINFLUX) of only the supercomplexes with two PSD95 molecules, since only these were used to calculate the average distance.

The 13 nm distance was measured using MINFLUX, as stated in the text. The fluorophore separation distances are shown in Figure 1dii.

(5) In the introduction, the authors could be more explicit in their explanation of memory formation and storage and how the presented study contributes to that field.

We thank the reviewer for the suggestion, but feel that such a discussion in the introduction would detract from the main points of the manuscript.

(6) Throughout the manuscript the authors prominently cite their own work, but relevant literature on synaptic plasticity and synapse nanostructure (EM and super-resolution studies) is lacking.

Further references have now been added.

(7) The results depicted in Figure 4b would be easier to interpret if a stacked histogram (including error bars) was used.

We agree that the data could be presented in such a way, but that would prevent the results from the biological repeats, along with the variation, being presented.